# The Exposome Perspective: Environmental and Infectious Agents as Drivers of Cancer Disparities in Low- and Middle-Income Countries

**DOI:** 10.3390/cancers17152537

**Published:** 2025-07-31

**Authors:** Zodwa Dlamini, Mohammed Alaouna, Tebogo Marutha, Zilungile Mkhize-Kwitshana, Langanani Mbodi, Nkhensani Chauke-Malinga, Thifhelimbil E. Luvhengo, Rahaba Marima, Rodney Hull, Amanda Skepu, Monde Ntwasa, Raquel Duarte, Botle Precious Damane, Benny Mosoane, Sikhumbuzo Mbatha, Boitumelo Phakathi, Moshawa Khaba, Ramakwana Christinah Chokwe, Jenny Edge, Zukile Mbita, Richard Khanyile, Thulo Molefi

**Affiliations:** 1SAMRC Precision Oncology Research Unit (PORU), DSI/NRF SARChI Chair in Precision Oncology and Cancer Prevention (POCP), Pan African Cancer Research Institute (PACRI), University of Pretoria, Hatfield 0002, South Africa; mohammed.alaouna@up.ac.za (M.A.); tebogo.marutha@up.ac.za (T.M.); rahaba.marima@up.ac.za (R.M.); rodney.hull@up.ac.za (R.H.); sikhumbuzo.mbatha@up.ac.za (S.M.); richard.khanyile@up.ac.za (R.K.); thulo.molefi@up.ac.za (T.M.); 2Wolfson Wohl Cancer Research Centre, School of Cancer Sciences, University of Glasgow, Garscube Estate, Switchback Road, Bearsden, Glasgow G61 1QH, UK; ntwasmm@unisa.ac.za; 3Department of Life and Consumer Sciences, College of Agriculture and Science & Environmental Sciences, Science Campus, University of South Africa, Florida 1709, South Africa; mkhizekwitshanaz@ukzn.ac.za; 4Department of Obstetrics and Gynecology, Gynecologic Oncology Unit, Charlotte Maxeke Johannesburg Academic Hospital, University of the Witwatersrand, Parktown, Johannesburg 2193, South Africa; langanani.mbodi@wits.ac.za; 5Department of Plastic, Reconstructive and Aesthetic Surgery, Steve Biko Academic Hospital University of Pretoria, Hatfield 0028, South Africa; nkhens@icloud.com; 6Department of Surgery, Charlotte Maxeke Johannesburg Academic Hospital, University of the Witwatersrand, Parktown, Johannesburg 2193, South Africa; thifhelimbilu.luvhengo@wits.ac.za; 7Next Generation Health, Division 1, Council for Scientific and Industrial Research, Meiring Naude Road, Brummeria, Pretoria 0001, South Africa; askepu@csir.co.za; 8Department of Internal Medicine, Division of Translational Research, School of Clinical Medicine, Faculty of Health Sciences, University of the Witwatersrand, 7 York Road, Parktown 2193, South Africa; raquel.duarte1@wits.ac.za; 9Department of Surgery, Faculty of Health Sciences, Steve Biko Academic Hospital, University of Pretoria, Hatfield 0028, South Africa; precious.setlai@up.ac.za; 10Department of Anatomical Pathology, National Health Laboratory Services, Tshwane Academic Division, University of Pretoria, Pretoria 0002, South Africa; benny.mosoane@nhls.ac.za; 11Department of Surgery, Nelson R Mandela School of Medicine, University of Kwa-Zulu Natal, 719 Umbilo Road, Durban 4001, South Africa; phakathib@ukzn.ac.za; 12 Department of Anatomical Pathology, Dr George Mukhari Academic Laboratory, National Health Laboratory Services, Sefako Makgatho Health Sciences University, Ga-Rankuwa 0208, South Africa; moshawa.khaba@nhls.ac.za; 13Department of Chemistry, College of Science, Engineering, and Technology, University of South Africa, Florida Park, Johannesburg 1709, South Africa; chokwrc@unisa.ac.za; 14Charlotte Maxeke Hospital, University of Witwatersrand, Johannesburg 2193, South Africa; jenny.edge@wits.ac.za; 15Department of Biochemistry, Microbiology and Biochemistry, University of Limpopo, Sovenga 0727, South Africa; zukile.mbita@ul.ac.za; 16Department of Medical Oncology, Faculty of Health Sciences, Steve Biko Academic Hospital, University of Pretoria, Hatfield 0028, South Africa

**Keywords:** cancer inequities, exposome, low- and middle-income countries (LMICs), environmental exposures, HIV-related cancers, oncogenic infections (HPV, HBV, and *H. pylori*), global health disparities, epigenetics

## Abstract

People in low- and middle-income countries face a higher risk of developing cancer owing to exposure to harmful environmental and infectious factors. These include polluted air and water, chemicals used in farming and industry, and infections such as HIV, hepatitis, and human papillomavirus. These exposures often occur together and can worsen their effects, especially when healthcare is difficult to access. This article looks at how a person’s total exposure to these harmful influences throughout their life, what scientists call the “exposome”, can help explain why cancer is more common and lethal in poorer countries. This study also explored how these risks affect individuals differently, based on their genes, diets, and living conditions. By understanding the full picture of exposure, this study shows how public health strategies can be improved. These include vaccination, early screening, safe work and home environments, and robust health systems. These findings are especially important for regions such as sub-Saharan Africa and South Asia, where cancer rates are increasing. This research helps point the way toward fairer and more effective methods for preventing and treating cancer worldwide.

## 1. Introduction

Cancer remains a leading cause of morbidity and mortality worldwide; however, its impact is not uniformly distributed. Low- and middle-income countries (LMICs) are projected to account for over 70% of the global cancer burden by 2040 [1]. This disparity stems not only from higher incidence rates but also from poorer survival outcomes, driven by challenges such as limited healthcare infrastructure, shortages of medical personnel, delayed diagnosis, and limited access to treatment [2].

Key drivers of cancer in LMICs include a higher prevalence of infectious agents, environmental pollutants, and occupational hazards, often due to insufficient regulatory oversight [3]. For example, hepatitis B virus (HBV) and human papillomavirus (HPV) are widespread in these regions and are major causes of liver and cervical cancer, respectively [4].

Neglected tropical diseases (NTDs), such as *Schistosoma haematobium*, *Opisthorchis viverrini*, and *Clonorchis sinensis*, are also associated with bladder and bile duct cancers [5]. These pathogens promote chronic inflammation and immune modulation, contributing to cancer development in the host [6]. Several helminths have been classified as Group 1 carcinogens by the International Agency for Research on Cancer (IARC), underscoring the role of infectious agents in cancer etiology [6].

In examining these and other environmental contributors to cancer risk, this review adopts the World Bank’s gross national income (GNI)-based classification to define LMICs (https://datahelpdesk.worldbank.org/knowledgebase/articles/906519-world-bank-country-and-lending-groups, accessed on 26 July 2025).

Within this context the exposome, a concept introduced by Christopher Wild in 2005, provides a comprehensive framework for understanding how non-genetic exposures, including environmental, infectious, and behavioral factors, interact with the genome over an individual’s lifetime [7,8]. This framework is particularly relevant for LMICs, where overlapping exposures to pollution, pathogens, and socioeconomic stressors create compounded cancer risks [9].

Although the exposome is traditionally used in individualized exposure science, this review adopts a population-level perspective. This manuscript investigates the cumulative exposures, such as industrial pollutants, occupational hazards, and endemic infectious diseases, that contribute to cancer disparities in LMICs. By evaluating how these exposures shape cancer outcomes, we highlight the relevance and potential of exposome-based methodologies, such as exposome-wide association studies (ExWASs), in advancing cancer research within LMIC contexts. The analysis explores disparities from both a global perspective, comparing LMICs to high-income countries in terms of cancer burden, healthcare access, and survival rates, and a localized view that addresses the geographic, economic, and social inequities within LMICs themselves. Through this dual lens, the manuscript underscores the urgency for context-specific research and policy interventions that incorporate exposomic insights to improve cancer prevention, diagnosis, and treatment in underserved populations.

## 2. The Exposome and Cancer Disparities

### 2.1. Definition and Components of the Exposome

The exposome encompasses the totality of environmental exposures that an individual encounters over the course of their life. These include both external factors, such as air and water pollution, occupational hazards, infections, and diet, and internal biological responses, including inflammation, oxidative stress, and metabolic changes (Figure 1) [9].

In the context of cancer, disparities in exposome profiles contribute significantly to the unequal burden of disease between LMICs and high-income countries (HICs). The interactions among environmental pollutants, infectious agents, and physiological stressors form a complex risk landscape that shapes cancer incidence and outcomes in LMICs [7,8].

#### 2.1.1. General External Exposure

In LMICs, widespread environmental pollutants such as fine particulate matter (PM2.5), heavy metals, and pesticides pose significant cancer risks. These exposures often result from unregulated industrial activity, agricultural practices, and weak environmental policies [10]. PM2.5, a common air pollutant, has been consistently associated with an elevated risk of respiratory cancers, particularly lung cancer [11].

Likewise, chronic exposure to heavy metals such as arsenic, lead, and cadmium, which are commonly found in contaminated water and soil, has been associated with an increased risk of lung, bladder, and skin cancers [12].

The use of agricultural pesticides also contributes significantly to environmental carcinogen exposure. Certain classes of pesticides, such as organochlorines and organophosphates, are classified as probable carcinogens and are associated with an increased risk of non-Hodgkin lymphoma and leukemia [10].

Climate change further amplifies these risks by creating environmental conditions that favor the proliferation of aflatoxin-producing fungi, particularly in warmer and more humid regions. Staple crops, such as maize and peanuts, are especially vulnerable. Aflatoxin exposure is a well-established risk factor for liver cancer, with the burden disproportionately affecting regions such as SSA and parts of Asia, where food safety regulations and monitoring are limited [13].

#### 2.1.2. Specific External Exposures

Specific external exposures such as infectious agents and dietary carcinogens are major contributors to cancer risk in LMICs. HPV is the leading cause of cervical cancer, accounting for approximately 70% of the global cases. In SSA, low HPV vaccination coverage and limited cervical cancer screening programs contribute to the disproportionately high incidence rates [14]. Similarly, chronic infections with HBV and *Helicobacter pylori* (*H. pylori*) are prevalent in LMICs and significantly elevate the risk of liver and gastric cancers, respectively, by promoting chronic inflammation and cellular damage in target organs [15,16]. Nutritional deficiencies and dietary exposures further exacerbate cancer risk. Economic constraints in LMICs often result in diets lacking essential micronutrients while containing high levels of carcinogens such as nitrosamines, which are formed during the processing of preserved meats. These substances are strongly associated with an increased risk of colorectal cancer [17].

#### 2.1.3. Internal Biological Responses

Internal biological responses, particularly chronic inflammation and metabolic dysregulation, are central mechanisms through which external exposures contribute to cancer development [18]. Chronic inflammation is a hallmark of persistent infection and environmental insults. For example, *H. pylori* infection induces sustained gastritis, creating a pro-tumorigenic environment that significantly increases the risk of gastric adenocarcinoma and, in some cases, primary gastric lymphoma [19].

In parallel, metabolic alterations, especially those associated with obesity, are emerging as critical risk factors for cancer in LMICs. Urbanization and dietary transitions toward energy-dense, nutrient-poor foods have driven rising obesity rates in these regions [20]. Obesity is associated with an increased risk of hormone-dependent malignancies, such as breast, ovarian, and endometrial cancers. This is largely due to obesity-induced metabolic disruptions, including insulin resistance and elevated circulating estrogen levels, which collectively promote tumor initiation and progression [21].

### 2.2. Role of the Exposome in Understanding Disparities

#### 2.2.1. Exposure Differences in LMICs Compared with High-Income Countries

LMICs experience a disproportionately high burden of environmental and infectious carcinogenic exposures compared to HICs [3]. A notable example is arsenic contamination in drinking water, particularly in parts of South Asia, where millions are chronically exposed to levels exceeding international safety standards. Prolonged arsenic exposure is associated with an increased risk of skin, lung, and bladder cancers. In contrast, HICs have largely mitigated this risk through stringent water quality regulations and remediation efforts [22].

Occupational exposure also varies significantly across regions. In LMICs, workers in sectors such as mining, construction, and agriculture are frequently exposed to carcinogens such as asbestos and benzene, which are linked to mesothelioma and leukemia, respectively [23]. While regulatory frameworks and occupational safety measures have substantially reduced such risks in HICs, similar protections are often lacking or poorly enforced in LMICs, leaving workers vulnerable to preventable exposures [24].

#### 2.2.2. Cumulative Impact of Environmental and Infectious Agents on Cancer Outcomes

The cumulative interaction between environmental and infectious exposures significantly amplifies cancer risk, particularly in LMICs, where regulatory and healthcare systems are often under-resourced. Co-exposure to aflatoxins and chronic HBV infection is a well-documented example that markedly increases the risk of hepatocellular carcinoma (HCC) due to their synergistic effects on DNA damage and liver inflammation. This phenomenon is especially prevalent in regions with inadequate food safety and storage practices.

Moreover, the convergence of environmental pollutants and oncogenic pathogens accelerates epigenetic alterations, such as aberrant DNA methylation. These changes can activate oncogenes and silence tumor suppressor genes, contributing to malignant transformation [25]. In LMICs, high levels of indoor air pollution primarily from the use of biomass fuel can compound these risks, particularly in the absence of effective healthcare infrastructure and preventive interventions. This interplay between multiple risk factors underscores the necessity of integrated cancer control strategies that go beyond addressing single exposures. Effective interventions must consider the broader environmental and social determinants of health to mitigate the disproportionate cancer burden in these settings [26].

## 3. Environmental Drivers of Cancer Disparities Within the Exposome Framework

Environmental drivers contributing to cancer disparities in LMICs, particularly across Africa, are complex and shaped by the region’s unique socioeconomic, ecological, and cultural contexts (Figure 2) [27]. The exposome framework, which captures the totality of environmental exposures over a lifetime, offers a valuable lens for understanding these disparities. One major contributor is indoor air pollution resulting from the widespread use of biomass fuels for cooking and heating, which is strongly associated with an increased risk of esophageal squamous cell carcinoma (ESCC) [28]. Occupational exposures to carcinogens, when combined with personal behaviors such as tobacco smoking, further elevate lung cancer risk in urban and industrial settings. Additionally, geographic variations in thyroid cancer prevalence have been observed in countries such as Tanzania, where the incidence appears to be linked to regional differences in dietary iodine levels, highlighting the interaction between environmental and nutritional factors [29]. These examples underscore how environmental risks vary not only between countries but also within them, reinforcing the need for localized interventions. Addressing these disparities requires context-specific public health strategies that incorporate local data on environmental exposure, infrastructure limitations, and sociocultural practices. Such tailored approaches are essential to effectively reduce the cancer burden in African LMICs [30].

### 3.1. Air and Water Pollution and Its Association with Cancer in LMICs

Air and water pollution are major environmental health challenges in LMICs, particularly in rapidly urbanizing regions of Africa (Figure 3). In cities such as Lagos (Nigeria) and Johannesburg (South Africa), rising industrial activity and traffic emissions have significantly degraded air quality. Common pollutants such as fine particulate matter (PM2.5), nitrogen dioxide (NO_2_), and volatile organic compounds (VOCs) are strongly associated with respiratory and urinary tract cancers, notably lung and bladder cancer [31].

Chronic NO_2_ exposure, in particular, correlates with elevated lung cancer incidence, although efforts to accurately assess this risk are often constrained by limited localized data [32]. In response, mitigation strategies such as the adoption of clean energy technologies, improved air quality monitoring, and sustainable urban planning are critical to reducing exposure and associated health risks [33].

Water pollution also contributes significantly to cancer risk, particularly in areas with poor sanitation and weak industrial waste regulations. In the Niger Delta, chronic oil pollution exposes communities to polycyclic aromatic hydrocarbons (PAHs) and heavy metals, such as arsenic and cadmium [34]. These contaminants induce oxidative stress and DNA damage, contributing to the development of liver, bladder, and gastric cancers (Figure 3). Addressing this requires investment in water quality monitoring, affordable filtration technologies, and stricter enforcement of environmental protection laws [34].

The Niger Delta exemplifies the complex link between environmental degradation, pollution-related exposure, and cancer risk, particularly where socioeconomic vulnerabilities amplify health impacts [35].

### 3.2. Occupational Exposure and Industrial Waste

Occupational exposure to carcinogens plays a significant role in cancer disparities across Africa, particularly in high-risk sectors such as mining and agriculture (Figure 4). In South Africa, gold miners face an elevated risk of lung cancer and mesothelioma due to their long-term exposure to silica dust and asbestos. The incidence of silicosis and related diseases, including lung cancer and tuberculosis, remains high among these workers [36].

Informal or artisanal mining further increases vulnerability due to poor safety measures and a lack of protective equipment [37]. Improper industrial waste management in rapidly urbanizing cities, such as Nairobi and Accra, exacerbates community-level exposure to environmental carcinogens [37]. The unregulated disposal of hazardous materials contaminates the air, water, and soil, increasing the risk of respiratory and gastrointestinal cancers. In South Africa, the legacy of asbestos mining continues to threaten surrounding communities with long-term exposure to asbestos-related diseases (ARDs) [37].

In the agricultural sector, the widespread use of pesticides, especially those containing carcinogenic compounds, has been linked to various hematologic malignancies. Limited regulation and poor training on safe pesticide use mean that many farm workers operate without protective equipment, increasing their cancer risk [37].

Addressing these occupational and environmental threats requires integrated policy approaches that strengthen workplace safety, enforce hazardous waste regulations, and promote sustainable industrial and agricultural practices [38].

### 3.3. Urbanization and Deforestation

Urbanization across African countries has introduced major public health challenges, particularly in terms of worsening air quality. In many low-income urban areas, reliance on biomass fuels such as charcoal and firewood for cooking, especially in poorly ventilated homes, contributes significantly to indoor air pollution, a known risk factor for respiratory diseases [39]. The promotion of cleaner cooking technologies and public awareness campaigns has begun to reduce this health burden [39]. Transitioning to cleaner fuels offers substantial public health benefits by lowering the incidence of pollution-related diseases [39].

Deforestation, which is largely driven by agricultural expansion and urban sprawl, undermines environmental and health systems. Forest loss contributes to greenhouse gas emissions and erodes natural buffers that are critical for climate regulation and food security [40]. These ecological disruptions increase drought frequency and food insecurity, indirectly weakening immune health and increasing disease vulnerability. Moreover, unplanned urban expansion often pushes informal settlements into industrial zones, where residents are exposed to air and water pollutants [41].

This overlap between environmental degradation and urban poverty amplifies the risk of cancer and other chronic diseases. The interlinked effects of urbanization, deforestation, and climate change pose a growing threat to public health in Africa. With urban areas expected to house over 50% of the continent’s population by 2050, there is an urgent need to address inefficient biomass fuel use, especially charcoal, which is four to six times less efficient than wood fuel. Community-based forest management and REDD+ initiatives offer viable solutions for reducing emissions while supporting sustainable livelihoods [41].

### 3.4. Lifestyle Factors Compounded by External Exposure

In African populations, lifestyle factors, including tobacco use, alcohol consumption, and changing dietary patterns, interact with environmental exposures to increase cancer risk. Tobacco smoking remains prevalent in many regions and significantly compounds the carcinogenic effects of air pollution. Combined exposure to smoking and airborne pollutants contributes to higher rates of lung, bladder, and oral cancers [42].

Alcohol consumption also increases the risk of cancer, particularly when combined with exposure to aflatoxins or infectious agents. A growing body of research links alcohol intake to breast cancer, with increasing consumption observed among women and youth in sub-Saharan Africa (SSA) due to shifting cultural norms and Western influence [43].

Urbanization has also altered dietary behaviors, leading to an increased intake of ultra-processed foods and red meat, which are associated with colorectal and obesity-related cancers. In contrast, traditional African diets rich in grains, legumes, and vegetables offer protective effects [44]. Promoting traditional diets through public health campaigns and urban agriculture may help counter rising cancer trends. However, in informal settlements, limited access to healthy food and persistent environmental exposure remain significant barriers. Specific regional dietary staples have also been associated with the risk of cancer. Mabusela et al. [45] identified a link between maize meal consumption and endemic esophageal squamous cell carcinoma. The authors found that esterified linoleic acid in maize meal degrades during storage, increasing prostaglandin E_2_ (PGE_2_) production and promoting low-acid reflux, which is a risk factor for esophageal cancer. Similarly, in the Eastern Cape, traditional beer consumption has been associated with esophageal cancer [46].

## 4. Infectious Agents as a Component of the Exposome

### 4.1. Disproportionate Prevalence of Oncogenic Infectious Agents in LMICs

#### 4.1.1. HPV and Cervical Cancer

HPV is the primary etiological agent of cervical cancer globally, with LMICs experiencing the highest disease burden. SSA, in particular, records some of the highest cervical cancer incidence and mortality rates due to limited access to HPV vaccination and organized screening programs [47]. Persistent infection with high-risk HPV genotypes, most notably HPV-16 and HPV-18, is strongly associated with the progression from cervical intraepithelial neoplasia to invasive cervical cancer [48]. Structural barriers such as inadequate healthcare infrastructure, logistical challenges in vaccine distribution, and cultural resistance to vaccination exacerbate the burden in LMICs [49]. Moreover, the prevalence of HPV and co-infection with multiple HPV types is significantly elevated in HIV-positive women, compounding cervical cancer risk in immunocompromised populations [50]. To address this, integrating HPV vaccination into national immunization schedules and school-based health programs has shown promise in improving vaccine uptake. Additionally, expanding access to affordable screening tools, such as HPV DNA testing and visual inspection with acetic acid (VIA), is essential [51]. The World Health Organization’s (WHO) global strategy to eliminate cervical cancer underscores the importance of achieving high HPV vaccination coverage, aiming to significantly reduce cervical cancer incidence and mortality in LMICs over the coming decades [52].

#### 4.1.2. HBV and HCV in Liver Cancer

Chronic infections with HBV and hepatitis C virus (HCV) are major drivers of hepatocellular carcinoma (HCC), especially in Africa and Asia. HBV often becomes chronic through perinatal transmission or early childhood exposure, significantly increasing the lifetime risk of liver cancer [53]. HCV is predominantly transmitted through blood exposure and poses a heightened risk in regions with inadequate blood screening and where intravenous drug use is common [54]. The combined exposure to viral hepatitis and aflatoxins, such as the naturally occurring mycotoxins found in improperly stored grains, further elevates HCC risk, particularly in LMICs [54]. This synergistic interaction highlights the need for integrated prevention strategies. Expanding neonatal HBV vaccination programs is a proven and cost-effective intervention that dramatically reduces the incidence of chronic HBV infection and subsequent HCC [55]. In addition, promoting food safety and proper grain storage is essential to limit exposure to aflatoxins. The advent of direct-acting antivirals (DAAs) has revolutionized HCV treatment, resulting in significant reductions in viral load, liver damage, and liver cancer incidence in treated populations [56].

#### 4.1.3. Role of HIV in Increasing Cancer Susceptibility

Human immunodeficiency virus (HIV) increases cancer susceptibility through immune suppression, which facilitates the persistence and proliferation of oncogenic viruses such as HPV, Epstein–Barr virus (EBV), and Kaposi’s sarcoma-associated herpesvirus (KSHV). Consequently, individuals with chronic HIV infection are at a heightened risk for AIDS-defining cancers, including Kaposi’s sarcoma, non-Hodgkin lymphoma, and cervical cancer, especially in high-prevalence regions such as Southern Africa [57]. While antiretroviral therapy (ART) has improved life expectancy and offers opportunities for integrated cancer care, access to ART and screening programs remains uneven across LMICs [58]. HIV-positive individuals often experience impaired cancer immune surveillance, contributing to delayed diagnoses and more advanced disease at presentation [59]. For instance, women living with HIV are less likely to undergo regular cervical cancer screening, exacerbating their risk [60]. Improving outcomes in HIV-affected populations requires better integration of HIV care with cancer screening and prevention services [61]. Moreover, the increased longevity of individuals on ART has been accompanied by a rise in non-AIDS-defining cancers (NADCs). This trend is driven by co-infection with oncogenic viruses and higher prevalence of other risk factors, such as smoking and alcohol use, among people living with HIV [62]. These patterns underscore the need for tailored cancer surveillance and early detection strategies in long-term ART recipients.

#### 4.1.4. Helicobacter Pylori Infection and Gastric Cancer

*H. pylori* is classified as a Group I carcinogen by the World Health Organization due to its strong association with chronic gastritis, peptic ulcers, gastric mucosa-associated lymphoid tissue (MALT) lymphoma, and gastric adenocarcinoma. The burden of *H. pylori*-associated gastric cancer is particularly high in North Africa, driven by poor sanitation, limited access to clean water, and dietary exposure to carcinogens such as nitrosamines [63,64]. Epidemiological studies estimate that approximately 80–90% of gastric cancer cases globally are linked to *H. pylori* infection, underscoring its pivotal role in gastric carcinogenesis [65]. The bacterium’s ability to induce chronic inflammation and alter gastric epithelial cell biology through virulence factors, such as CagA and VacA, contributes to this elevated cancer risk. Effective public health interventions, including improved sanitation, population-level screening, and access to eradication therapies, are essential for reducing the incidence of *H. pylori*-related gastric cancer [66]. Additionally, dietary factors, particularly the intake of antioxidants such as vitamins C and E, have been shown to modulate infection risk and disease progression, emphasizing the importance of integrative strategies that address both environmental and nutritional determinants [67].

### 4.2. Neglected Tropical Diseases (NTDSs) and Cancer

Neglected tropical diseases (NTDs) primarily affect the poorest and most marginalized communities, specifically those without access to clean water, adequate sanitation, and proper housing. Poor hygiene and overcrowded conditions increase the vulnerability to infection, especially in the tropical and subtropical regions. Globally, approximately 1.5 billion people (24% of the world’s population) are infected with soil-transmitted helminths, with the highest prevalence in SSA, China, South America, and Southeast Asia [51].

Environmental conditions, such as warm climates, poor sanitation, and limited access to clean water, facilitate the continuous transmission of these infections. Even after treatment, reinfection is common owing to persistent environmental exposure [68]. Helminths have evolved sophisticated mechanisms to evade the immune system, including immune modulation, which inadvertently impairs the host’s ability to combat other pathogens [69]. This immunomodulation has been implicated in decreased cancer immune surveillance and the promotion of chronic inflammation, both of which increase oncogenic risk [70,71,72].

Infection with *O. viverrini* and *C. sinensis* is strongly linked to cholangiocarcinoma (bile duct cancer) [73], whereas *S. haematobium* is responsible for 46–75% of bladder cancers in endemic regions [74]. In addition, *Schistosoma japonicum* has been associated with colorectal cancer due to chronic inflammation caused by egg deposition [74]. Similarly, *Schistosoma mansoni* activates hepatocellular carcinoma-associated transcription factors (c-Jun and STAT3) in liver cells, promoting hepatocarcinogenesis [75]. Even protozoa previously considered benign, such as *Blastocystis hominis*, have been associated with colorectal cancer [76]. These findings underscore the strong etiological link between NTDs and cancer, particularly in LMICs, where both conditions are highly endemic and often underdiagnosed.

### 4.3. Interaction of Infectious Agents with Other Environmental Exposures

The interplay between infectious agents and environmental exposure is a critical contributor to cancer development, particularly in LMICs. When combined, these factors often amplify each other’s carcinogenic potential, increasing the risk of cancer through synergistic mechanisms of action. A well-documented example is co-exposure to HBV and aflatoxins, which significantly elevates the risk of HCC. Aflatoxins promote DNA adduct formation, whereas HBV suppresses immune surveillance, creating a permissive environment for cellular transformation [77]. Similarly, exposure to heavy metals can intensify the oxidative stress and DNA damage caused by *H. pylori* [78]. This combination leads to chronic inflammation, a hallmark of carcinogenesis, and increases the risk of gastric cancer [79].

HIV-induced immunosuppression also increases cancer susceptibility by facilitating persistent infections with oncogenic viruses such as HPV and Epstein–Barr virus. This immune compromise contributes to higher rates of cervical cancer and Kaposi’s sarcoma [80]. Environmental pollutants, including particulate matter and waterborne toxins such as arsenic, can further impair DNA repair mechanisms, thereby increasing the cancer risk in individuals already infected with oncogenic pathogens.

To mitigate these cumulative risks, public health interventions must address both infectious and environmental factors. Priorities include expanding HBV vaccination, improving grain storage to prevent aflatoxin contamination, implementing *H. pylori* eradication strategies, and enforcing industrial regulations to reduce air and water pollution. Enhanced environmental surveillance is crucial for monitoring and managing these overlapping exposures (Figure 5).

### 4.4. Case Studies: Regions with a High Burden of Infection-Associated Cancer

#### 4.4.1. East Africa

Cervical cancer remains a major public health concern in East Africa, particularly in countries such as Uganda and Kenya, where it is the leading cause of cancer-related deaths among women. The region faces alarmingly low HPV vaccination coverage, with rates reported to be less than 30%. This, combined with limited access to cervical cancer screening, contributes to persistently high mortality rates [81].

Recent studies have indicated that vaccine uptake is especially low in urban and peri-urban settings, with coverage as low as 8.6% in Kampala and 19.6% in Lira City, Northern Uganda [82]. Contributing factors include inadequate awareness of HPV and cervical cancer, widespread cultural misconceptions, and systemic barriers to vaccine distribution [83]. Integrating HPV vaccination into school-based health programs is a promising strategy to overcome these challenges. Such programs have been associated with improved vaccine acceptance and increased coverage among adolescent girls, helping reduce both the incidence and mortality of cervical cancer [84].

Moreover, expanding access to cost-effective diagnostic methods, such as HPV DNA testing, is essential for scaling up cervical cancer screening. The WHO advocates a comprehensive three-pronged approach that includes HPV vaccination, routine screening, and prompt treatment of precancerous lesions (Figure 6) [85].

#### 4.4.2. West Africa

In West Africa, particularly in Nigeria and Ghana, liver cancer remains a major public health concern, largely driven by chronic HBV infection and environmental exposure to aflatoxins (Figure 6). Aflatoxin B1 (AFB1), a toxin produced by molds that commonly contaminate food crops such as maize and ground nuts, is strongly associated with HCC, especially in individuals co-infected with HBV [86]. A systematic review by Liu et al. (2012) confirmed that the synergistic effect of AFB1 and chronic HBV infection significantly elevates the risk of developing liver cancer in regions where both exposures are prevalent [87]. Cultural practices such as the use of shared razors and traditional scarification also contribute to HBV transmission, highlighting the importance of culturally sensitive community education [88]. To address these multifactorial risks, public health strategies should focus on improving neonatal HBV vaccination coverage, which has been proven to reduce transmission and the subsequent development of liver disease. Comprehensive interventions that combine vaccination with educational campaigns on food safety and infection prevention are essential to reduce the liver cancer burden in this region [88].

#### 4.4.3. Southern Africa

Southern Africa faces a substantial public health burden driven by the intersection of high HIV prevalence and AIDS-defining cancers (ADCs), particularly Kaposi’s sarcoma (KS) and non-Hodgkin lymphoma (NHL) (Figure 6). These cancers are significantly more prevalent in regions with high HIV infection rates. The introduction of highly active antiretroviral therapy (HAART) has led to a decline in the incidence and mortality rates of KS and NHL [89]. Persistent inequities in access to antiretroviral therapy (ART) continue to negatively impact cancer outcomes in the region [90].

Integrating cancer care into existing HIV treatment frameworks is a critical strategy for improving outcomes, enabling concurrent management of both conditions [91]. Additionally, HIV-positive women are at an increased risk of cervical cancer, necessitating the incorporation of cervical cancer screening into HIV care protocols. Evidence suggests that such integrated models improve screening uptake and facilitate earlier diagnosis, which is key to effective treatment [92].

#### 4.4.4. North Africa

In North Africa, particularly Egypt, the prevalence of gastric cancer is closely linked to high rates of *H. pylori* infection and associated dietary risk factors (Figure 6). *H. pylori* is a well-established etiological agent in gastric carcinogenesis, contributing to chronic gastritis, peptic ulcers, and ultimately gastric adenocarcinoma [93]. The World Health Organization classifies *H. pylori* as a Group 1 carcinogen due to its strong association with gastric cancer [94]. Genetic susceptibility further exacerbates this risk. For example, individuals with the GSTM1 null genotype are more vulnerable to gastric cancer when infected with *H. pylori* [95].

Inadequate sanitation and limited access to effective *H. pylori* eradication therapies in the region exacerbate this public health challenge. Many affected individuals remain undiagnosed and untreated, increasing the risk of advanced disease progression. To address this issue, national efforts should focus on improving sanitation infrastructure and expanding access to affordable diagnostic and treatment options. Implementing population-wide *H. pylori* screening and eradication programs could enable the early detection and prevention of gastric cancer, thereby reducing the disease burden across the region [96].

## 5. Mechanistic Pathways Linking the Exposome to Cancer

### 5.1. Synergistic Effects of Environmental Toxins and Infectious Agents

The exposome plays a pivotal role in determining cancer risk [7]. Understanding the mechanistic pathways through which these exposures exert their carcinogenic effects is crucial for identifying prevention strategies (Figure 7), particularly in LMICs, where the burden of environmental toxins and infectious diseases is often higher than that in HICs. This interplay between these two factors underscores the complex, multifactorial etiology of cancer, with environmental and infectious agents acting together to amplify the risk and progression of cancer. Viral infections inhibit DNA repair mechanisms, facilitating the integration of viral genomes into the host cell’s DNA. This integration disrupts cellular homeostasis and inhibits apoptosis, allowing the persistence of chronic infections. Over time, these chronic infections give rise to precancerous cells characterized by compromised tumor suppressor gene function and dysregulation of cell cycle genes. These alterations lead to aberrant epigenetic modifications that further exacerbate genomic instability. The resulting genomic instability and persistent inflammation create a microenvironment favorable for tumor development and progression, ultimately culminating in cancer (Figure 7).

#### 5.1.1. Epigenetic Changes Driven by Chronic Exposure

Epigenetic modifications, such as DNA methylation, histone changes, and altered non-coding RNA expression, are key mechanisms through which environmental toxins and infectious agents contribute to cancer development [97].

These heritable changes do not alter the DNA sequence but influence gene expression, often silencing tumor suppressor genes or activating oncogenes, thereby promoting tumorigenesis [98]. Chronic exposure to environmental toxins, including heavy metals, pesticides, and air pollutants, induces epigenetic changes. For instance, long-term As exposure is linked to skin, lung, and bladder cancers through hypermethylation of tumor suppressor genes [98]. These modifications disrupt normal cellular regulation and facilitate malignant transformation [99].

Infectious agents such as HPV and HBV also play a prominent role in reshaping the epigenetic landscape. Persistent infection with high-risk HPV strains leads to viral DNA integration and deregulation of cell cycle genes via epigenetic changes [100]. Similarly, HBV infection, particularly in endemic areas such as SSA and East Asia, has been associated with abnormal DNA methylation of cancer-related genes, such as *cyclin-dependent kinase inhibitor 2A (CDKN2A)*, commonly known as *p16INK4a*, and Ras association domain family member 1 (*RASSF1A*), both of which play critical roles in liver carcinogenesis [101].

Importantly, co-exposure to environmental carcinogens and infectious agents may have synergistic effects, enhancing epigenetic disruption and amplifying cancer risk [9]. This convergence illustrates the complex multilevel interactions between the exposome and cancer pathways in vulnerable populations.

#### 5.1.2. Oxidative Stress and DNA Damage Pathways

Reactive oxygen species (ROS), which are natural byproducts of cellular metabolism, can damage DNA, promote mutations, and alter the signaling pathways that drive tumorigenesis. Chronic exposure to environmental pollutants such as fine particulate matter (PM2.5), benzene, and pesticides elevates ROS production, leading to oxidative DNA damage, genomic instability, and ultimately, cancer [102]. For instance, benzene exposure is strongly associated with increased ROS generation, resulting in DNA strand breaks and mutations that contribute to leukemia [103]. Infections also play a significant role: *H. pylori* causes chronic gastric inflammation and promotes the release of ROS and reactive nitrogen species that damage DNA and contribute to gastric carcinogenesis [19]. Similarly, persistent HPV infection activates oxidative stress pathways that, in conjunction with viral oncogene expression and pre-existing genetic alterations, contribute to cervical cancer development [104]. The synergistic effect of environmental pollutants and infectious agents further amplifies oxidative stress and causes DNA damage. This dual burden is particularly pronounced in LMICs, where co-exposure to both sources is common and significantly elevates the risk of cancer.

### 5.2. Immune System Dysregulation and Chronic Inflammation

#### Persistent Infection and Cancer Risks

Persistent infection with high-risk strains of HPV is a major cause of cervical cancer. It triggers chronic inflammation in the cervical epithelium, characterized by sustained immune activation and the release of pro-inflammatory cytokines by macrophages and T lymphocytes. This inflammatory microenvironment promotes cellular transformation and tumor development [86]. Similarly, chronic hepatitis B virus (HBV) infection induces long-term hepatic inflammation. Continuous immune cell activation leads to elevated levels of cytokines such as tumor necrosis factor-alpha (TNF-α) and interleukin-6 (IL-6), which support the survival and proliferation of infected hepatocytes [105]. These cytokines also contribute to genomic instability, which is a key mechanism of HCC development. Tumor-promoting inflammation caused by chronic infections facilitates immune evasion and angiogenesis, further enhancing cancer progression (Figure 7) [105].

### 5.3. Molecular and Genetic Susceptibilities Influenced by Cumulative Exposures

#### 5.3.1. Genetic Polymorphisms and Susceptibility to Environmental Toxins

Genetic polymorphisms significantly influence an individual’s ability to metabolize and detoxify environmental toxins, thereby modifying their susceptibility to environmentally driven cancers [106]. Variants in genes involved in xenobiotic metabolism and DNA repair can either mitigate or exacerbate the carcinogenic effects of exposures. Cytochrome P450 enzymes (CYPs), particularly *CYP1A1*, play a central role in the metabolism of polycyclic aromatic hydrocarbons (PAHs) found in tobacco smoke and industrial pollutants. Polymorphisms in *CYP1A1* are associated with increased lung cancer risk, especially among smokers [107]. Similarly, deletions or functional variants in *GSTM1* and *GSTT1*, which encode glutathione S-transferase enzymes essential for detoxifying reactive intermediates, are associated with an elevated risk of bladder and lung cancers in individuals exposed to environmental toxins [108]. Variants in *NQO1*, an enzyme involved in quinone detoxification, are also correlated with a heightened cancer risk in settings of high air pollution [109]. In addition to metabolic genes, emerging evidence implicates polymorphisms in circadian rhythm genes such as *ARNTL*, *NPAS2*, and *RORA* in cancer susceptibility. Disruptions in circadian cycles due to night shift work are increasingly recognized as carcinogenic factors. Individuals with specific variants in these genes show increased risk for aggressive prostate cancer and potentially other malignancies, particularly when exposed to chronic circadian disruption [110]. These findings underscore the critical role of gene–environment interactions in the development of cancer. Identifying high-risk genetic profiles could support more personalized cancer prevention strategies, particularly in populations facing sustained environmental exposures.

#### 5.3.2. Gene–Environment Interactions in Infectious-Related Cancer Risk

Gene–environment interactions play a pivotal role in modulating cancer risk associated with chronic infections, particularly in regions where oncogenic pathogens such as HPV, HBV, and *H. pylori* are highly prevalent [111]. Host genetic variations that influence immune response, inflammation, and infection clearance can significantly modify cancer susceptibility in infected individuals [108].

Polymorphisms in the human leukocyte antigen (HLA) system, particularly in *HLA-DQ* alleles, have been linked to an increased risk of cervical cancer in women with persistent HPV infection [112]. These HLA variants may compromise antigen presentation and impair immune-mediated viral clearance, allowing for prolonged HPV persistence and increasing the likelihood of progression to malignancy [113]. Similarly, single nucleotide polymorphisms (SNPs) in immune-regulatory genes such as *IL-10* have been associated with elevated HCC risk in individuals with chronic HBV infection [114]. Reduced IL-10 expression may result in insufficient control of hepatic inflammation, promoting fibrosis and carcinogenesis. In the context of *H. pylori* infection, polymorphisms in pro-inflammatory cytokine genes, including *IL-1β* and *TNF-α*, can intensify mucosal inflammation, leading to chronic gastritis and heightened risk of gastric cancer [16]. Collectively, these examples highlight how genetic predispositions, especially those affecting immune and inflammatory pathways, can magnify the oncogenic potential of chronic infections. Understanding these interactions is essential for developing targeted prevention and screening strategies in high-risk populations.

#### 5.3.3. Cumulative Exposures and Their Impact on Genetic Stability

Chronic exposure to environmental toxins such as tobacco smoke, air pollution, and industrial chemicals induces a range of DNA lesions, including base modifications, strand breaks, and crosslinks. If unrepaired, these lesions can accumulate and initiate carcinogenesis [115]. Persistent exposure can overwhelm cellular DNA repair systems, resulting in genomic instability and an increased mutation burden. The situation is exacerbated when combined with chronic infections like HPV or HBV, which further compromise genomic integrity through viral DNA integration into the host genome [113]. This cumulative DNA damage significantly increases the risk of malignant transformation. Moreover, genetic predispositions that affect DNA repair efficiency can amplify these risks. Polymorphisms in key repair genes, such as *TP53* and *BRCA1/BRCA2*, impair DNA damage correction, facilitating mutation accumulation. For instance, individuals with *BRCA1* mutations, which are critical for repairing double-strand breaks, are at a markedly elevated risk of breast and ovarian cancers [116].

### 5.4. Cultural Practices and Cancer Risk in LMICs

Cultural traditions and social behaviors significantly influence cancer risk in LMICs. Practices embedded in daily life—shaped by socioeconomic conditions, traditional beliefs, and limited health education—contribute to the burden of cancers, especially those affecting the lungs, esophagus, and liver [117]. Although ethnicity, population origin, and regional diversity affect cancer patterns in SSA, the role of cultural attitudes and traditional medicine in cancer prevention and care remains underexplored [117].

One notable practice is the use of traditional tobacco pipes, particularly by women in Southern Africa. Unlike filtered cigarettes, pipe tobacco involves deep inhalation of unfiltered smoke, increasing exposure to carcinogens and elevating the risk of lung, oral, and esophageal cancers [118]. Cultural ceremonies in some Nguni communities also incorporate different types of tobacco, complicating efforts to introduce anti-smoking campaigns without addressing deeply rooted traditions [119]. Alcohol consumption, especially of unregulated sorghum-brewed beverages commonly used in community celebrations, has also been linked to an increased incidence of esophageal and liver cancers. These homemade brews may contain harmful contaminants that exacerbate their carcinogenic potential [120].

Urbanization has driven shifts in lifestyle, including increased sedentary behavior and dietary change. The replacement of traditional, high-fiber, plant-based diets with ultra-processed foods has contributed to rising obesity rates, which are associated with breast, colorectal, and endometrial cancers [121]. Additionally, rural and informal communities often lack access to health education, leading to misconceptions regarding cancer etiology and symptoms. Cultural norms, gender roles, and daily hardships—such as walking long distances for water or food—can delay health-seeking behavior, often resulting in late-stage diagnoses [122].

Culturally tailored public health strategies are essential for addressing these challenges. Interventions that engage community leaders, respect local customs, and provide accessible education can encourage healthier behavior. Integrating cancer prevention into culturally relevant platforms and improving community health literacy are critical steps toward reducing culturally mediated cancer risk.

## 6. Mitigating Cancer Disparities Through an Exposome-Informed Approach

### 6.1. Prevention Strategies

#### 6.1.1. Vaccination Campaigns

Vaccination remains one of the most effective strategies for preventing cancers linked to infectious agents, particularly HPV and HBV. These viruses are major contributors to cervical and liver cancers, respectively, and are disproportionately prevalent in LMICs, exacerbating cancer disparities [123]. The global burden of cervical cancer, primarily driven by persistent HPV infection, is heavily concentrated in LMICs, where HPV vaccination coverage is low. However, there have been notable successes. For example, Rwanda’s national HPV vaccination program has achieved over 90% coverage among adolescent girls, resulting in a measurable decline in cervical cancer incidence [124].

Despite this potential, the widespread implementation of vaccination programs in LMICs faces multiple barriers. These include high vaccine costs, limited infrastructure for delivery, logistical challenges, and cultural resistance to immunization, often requiring community-specific education and outreach strategies [125]. Similarly, Ghana’s introduction of a national HBV vaccination program in the early 2000s has contributed to a decline in chronic HBV infection, a leading cause of liver cancer in the region [126]. Globally, universal HBV vaccination has been significantly successful in reducing liver cancer incidence, as observed in Taiwan and China. However, in many LMICs, the HBV vaccination coverage remains suboptimal. Strengthening vaccine supply chains, improving public awareness of HBV’s oncogenic potential, and addressing socioeconomic barriers are essential steps to expanding vaccine uptake in high-risk regions [127].

#### 6.1.2. Environmental Policy Reforms

Environmental exposures such as air pollution, heavy metals, and contaminated water are major contributors to cancer risk in LMICs. Implementing robust environmental policies to reduce these exposures is crucial for cancer prevention, particularly in urban and industrial regions [3].

Air pollution, especially fine PM2.5, is a well-established carcinogen linked to lung cancer, cardiovascular disease, and respiratory illnesses. The Global Burden of Disease Study attributes a substantial portion of cancer-related deaths in LMICs to PM2.5 exposure [128]. Cities like Lagos (Nigeria) and Cairo (Egypt) experience high levels of pollution due to industrial emissions, vehicular exhaust, and unregulated waste burning [129]. To address this, governments must enforce stricter air quality standards, invest in clean energy solutions, and incentivize the adoption of low-emission technologies. The African Development Bank’s initiatives to support clean energy and green infrastructure represent an important step toward mitigating environmental carcinogens [130].

Water contamination is a pressing concern, especially in countries such as Nigeria, Ghana, and Burkina Faso, where high levels of arsenic in groundwater pose a serious public health threat, increasing the risk of skin, lung, and bladder cancers [131]. In response, both national governments and international partners are investing in solutions such as improved purification systems, community-based water filtration, and affordable water testing technologies to ensure access to safe drinking water [132].

### 6.2. Advancing Exposomic Research in LMICs

#### 6.2.1. Affordable and Accessible Technologies for Exposure Assessment

Accurate and affordable exposure assessment tools are critical for understanding the health impacts of environmental hazards in LMICs [133]. The growing availability of low-cost sensors has enabled real-time monitoring of air pollutants such as PM2.5, NO_2_, and sulfur dioxide (SO_2_). These tools support policymakers in identifying high-risk areas and implementing targeted interventions [134]. Air quality monitoring is gaining attention in African cities in response to increasing urbanization and industrial growth. Organizations such as the African Network for the Prevention of Occupational and Environmental Cancer (ANPOEC) have promoted the deployment of low-cost sensors to monitor air pollution and particulate matter, helping local governments develop data-driven policies to reduce exposure to carcinogens [135].

Water safety is another priority in many LMICs, where centralized infrastructure is often absent. Portable water-testing kits allow for the rapid detection of contaminants, such as arsenic, fluoride, and pathogens, in drinking water. These low-cost tools have empowered local health workers and communities to identify waterborne carcinogens and mitigate exposure risks [136].

Mobile phone applications and wearable devices are increasingly used in exposomic research and public health. These technologies can monitor individual exposure to pollutants in real time, thereby enabling informed health decisions. A study in India demonstrated the utility of mobile apps for tracking PM2.5 and educating users on pollution risks, highlighting the potential for similar tools in African settings [137].

#### 6.2.2. Integrating Exposomic Data into Public Health Interventions

To reduce cancer disparities, the systematic incorporation of exposomic data into public health strategies is essential. In African contexts where environmental, dietary, and infectious exposures intersect, this integration is essential for designing effective, context-specific cancer control programs [138]. For example, in Kenya, researchers have linked high liver cancer rates to overlapping HBV infection and aflatoxin-contaminated maize. The integration of such data has enabled coordinated interventions, including HBV vaccination and improved food safety [139]. In South Africa, combining air pollution data with cancer incidence rates revealed urban hotspots for lung cancer, informing the development of targeted screening and prevention efforts [140].

Beyond targeted interventions, the integration of exposomic data into public health frameworks can contribute to the overall strengthening of healthcare systems. By systematically monitoring environmental exposures across regions and populations, health authorities can identify emergent risks early, allocate resources more efficiently, and enhance surveillance capacity. Exposomic approaches also support a shift toward precision public health, where prevention, diagnostics, and treatment strategies are adapted to specific population-level exposure profiles. This is especially critical in resource-limited settings, where optimizing health system responses can maximize impact with limited infrastructure. Moreover, the inclusion of exposome-informed tools in health information systems promotes interdisciplinary collaboration between environmental scientists, epidemiologists, clinicians, and policymakers, fostering a more integrated and proactive approach to health system governance. These examples illustrate how exposomic insights can guide the development of more precise, evidence-based public health policies in LMICs.

### 6.3. Strengthening Healthcare Systems

#### Addressing Healthcare Access Inequities in LMICs

Access to cancer care in many African countries remains limited, particularly in rural and underserved regions of the continent. A major contributing factor is the shortage of cancer treatment facilities and diagnostic services. Strengthening local healthcare infrastructure through the establishment of regional cancer centers and the expansion of affordable diagnostic tools is a crucial step toward equitable care delivery [2]. The African Cancer Organization has called for broader access to cancer treatment services, emphasizing affordability and outreach to marginalized communities (African Cancer Organization, n.d.). In addition, international partnerships such as the collaboration between the Global Health Initiative (GHI) and the African Union (AU) have played a critical role in providing technical support, resources, and capacity-building programs to improve cancer care across the continent. These efforts have contributed to the development of regional cancer centers and enhanced the ability of local healthcare providers to deliver timely and effective treatment [141].

### 6.4. Policy and Global Collaboration

#### 6.4.1. International Partnerships for Exposome-Informed Cancer Control

International partnerships are pivotal in advancing exposome-informed cancer control in Africa. Collaborations with institutions such as the International Agency for Research on Cancer (IARC) have enhanced cancer surveillance, research infrastructure, and national control programs across the continent [119]. In countries such as Uganda and Kenya, joint efforts with the IARC have enabled the integration of exposomic research into public health strategies. Studies on environmental carcinogens, including aflatoxins and air pollution, have yielded critical insights into region-specific cancer risks in SSA [142]. These partnerships not only strengthen technical capacity but also promote data-driven interventions tailored to the environmental and infectious disease profiles of African nations.

#### 6.4.2. Funding and Capacity-Building for LMIC Research Infrastructure

Sustainable funding is critical for strengthening cancer research infrastructure in LMICs, particularly in Africa, where environmental exposures significantly contribute to cancer disparities [2]. International organizations, including the Wellcome Trust and the Bill & Melinda Gates Foundation, have supported research initiatives aimed at elucidating the links between environmental risk factors and cancer outcomes [143]. Simultaneously, capacity-building efforts have focused on training local scientists, healthcare professionals, and public health officials to conduct exposomic and molecular epidemiology research. Initiatives like the African Cancer Research Consortium (ACRC) provide funding, technical resources, and collaborative networks to empower regional research institutions [144]. These investments are essential for enabling context-specific cancer control strategies that incorporate environmental, infectious, and social determinants of health.

## 7. Case Studies and Success Stories

### 7.1. Community-Driven Cancer Prevention Programs Integrate Environmental and Infectious Risk Mitigation

Community-led cancer prevention initiatives have been substantially successful in reducing cancer risk, particularly in settings burdened by both environmental and infectious exposures. In Nigeria, farmer cooperatives have adopted hermetic grain storage methods to reduce aflatoxin contamination in staple crops, significantly lowering liver cancer risk associated with chronic aflatoxin exposure [145]. Community health workers (CHWs) play a vital role in raising awareness and promoting preventive behavior. In Uganda, CHWs have improved cervical cancer screening uptake through culturally tailored educational outreach [146]. In Cambodia, participatory health workshops that integrate clean water access with *H. pylori* education have improved sanitation practices and reduced gastric cancer incidence [147].

In urban Kenya, community leaders have launched grassroots campaigns to tackle pollution by establishing local recycling centers and clean water kiosks. These programs not only reduce environmental hazards but also strengthen community awareness of the links between pollution and cancer risk [148]. These examples underscore the effectiveness of community-based cancer prevention strategies. By leveraging local knowledge and engagement, such programs can simultaneously address environmental and infectious risks while fostering sustainable public health outcomes [149].

### 7.2. Success of Vaccination and Environmental Cleanup Efforts to Reduce Cancer Burden

Vaccination campaigns and environmental remediation efforts have significantly reduced the cancer burden in LMICs. In Rwanda, a national HPV vaccination program achieving over 90% coverage among adolescent girls has been instrumental in decreasing cervical cancer incidence, especially when paired with routine screening initiatives [150]. Similarly, Ghana’s infant HBV vaccination program has led to a marked decline in chronic HBV infections, a major driver of liver cancer in the region [49]. These programs have been further bolstered by culturally sensitive public education efforts that improve vaccine uptake by dispelling misinformation and promoting trust [151].

Environmental cleanup initiatives complement public health gains. In Nigeria’s Niger Delta, community-driven efforts to remediate oil-contaminated sites have reduced population-level exposure to carcinogenic PAHs [152]. In South Africa, asbestos decontamination projects in former mining communities have substantially lowered the incidence of mesothelioma [153]. Together, these cases underscore the power of integrated approaches combining vaccination, environmental restoration, and public education to address the root causes of cancer disparities in LMICs.

## 8. Future Directions and Research Priorities

### 8.1. Development of Exposome-Wide Association Studies (ExWASs) in LMICs

Exposome-wide association studies (ExWASs) provide a comprehensive framework for evaluating the cumulative effects of environmental, genetic, and infectious factors on cancer risk [9]. In LMICs, where the cancer burden is shaped by unique exposure profiles and socioeconomic constraints, ExWAS methodologies offer a critical path toward contextualized cancer prevention strategies. However, the implementation of ExWASs in these settings is hindered by the limited infrastructure for large-scale environmental exposure monitoring. While initiatives like the African Genome Variation Project have advanced genetic research, the integration of genomic data with high-resolution environmental exposure tracking remains insufficient [154]. A major research priority is the investigation of region-specific co-exposure. For instance, aflatoxin exposure prevalent in SSA due to poor grain storage synergizes with chronic HBV infection to significantly increase HCC risk [155]. Similarly, the interplay between urban air pollution, changing diets, and HPV infection warrants investigation for its role in cervical and colorectal cancer risk, particularly in rapidly urbanizing LMIC contexts [156].

To make ExWASs feasible and impactful in LMICs, capacity building is essential. Training local researchers in molecular epidemiology, biostatistics, and bioinformatics will enable context-relevant study design and data interpretation. International collaborations can support these efforts through resource sharing, mentorship, and co-development of research agendas aligned with regional cancer priorities (Figure 8) [157].

### 8.2. AI and Big Data Are Leveraged to Map Cumulative Exposures and Predict Cancer Risk

Artificial intelligence (AI) and big data analytics are transforming cancer research by enabling the integration of large, heterogeneous datasets to assess cumulative exposures and predict cancer risk [158]. AI-driven tools can analyze satellite imagery and environmental monitoring data to identify pollution hotspots, such as areas with elevated air or water contamination, that contribute to cancer incidence [159]. Machine learning algorithms, trained on population-specific data, can forecast cancer risk patterns, helping to guide targeted public health interventions [160].

Beyond risk prediction, AI has shown promise in community-based surveillance. AI-powered mobile health platforms enable real-time reporting of environmental exposures and symptoms by community members, thereby enhancing data collection and awareness [161]. These technologies support timely, locally tailored responses and strengthen community engagement in cancer prevention efforts [162].

However, infrastructure limitations and data scarcity, particularly in LMICs, remain significant barriers. Centralized, African-specific data repositories are urgently needed to support algorithm development and reduce bias from non-representative training datasets (Figure 8) [163]. Establishing standardized benchmarking systems is essential to ensure AI tools are both effective and equitable across diverse populations [160].

## 9. Conclusions

The exposome, a comprehensive measure of all environmental exposures across an individual’s lifetime, offers a powerful framework for understanding cancer disparities, particularly in LMICs [9]. In these regions, the convergence of environmental pollutants, infectious diseases, inadequate healthcare infrastructure, and socioeconomic inequalities intensifies cancer risk. By adopting an exposomic perspective, cancer research can move beyond isolated genetic or behavioral risk factors to embrace a more integrated view of disease causation. This holistic lens is especially critical for SSA and other LMICs, where exposures such as air and water pollution, aflatoxin-contaminated food, and persistent infections like HPV and HBV contribute significantly to the cancer burden [164]. Incorporating exposomic data into public health systems has the potential to enhance cancer prevention, early detection, and treatment, particularly in contexts where late-stage diagnoses are common and healthcare resources are limited [144]. Strengthening HPV and HBV vaccination programs can substantially reduce the incidence of cervical and liver cancers, while policies targeting environmental carcinogens could further alleviate preventable cancer cases [165]. Success depends on robust infrastructure, data integration, and equitable access to preventive services.

Realizing the promise of exposomics requires global collaboration, sustained investment, and capacity building in LMICs. Integrating exposomic research into national cancer control strategies can drive more equitable health outcomes. Ultimately, a comprehensive, exposome-informed approach can reshape global cancer prevention, making it more inclusive, context-specific, and effective across all populations.

## Figures and Tables

**Figure 1 cancers-17-02537-f001:**
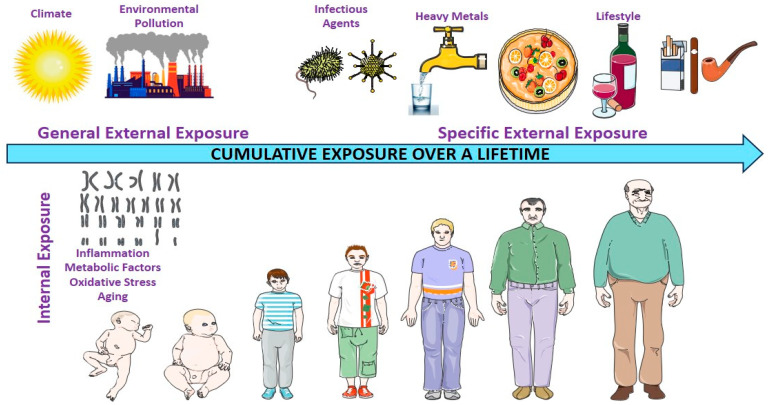
Environmental exposure experienced by individuals throughout their lifetime. This figure illustrates the exposome framework, comprising three interconnected domains that capture the totality of environmental exposure throughout an individual’s lifespan. The general external domain encompasses broad contextual factors, including climate, urban or rural living environments, socioeconomic status, and ambient pollution (e.g., air quality and industrial emissions). The specific external domain includes more targeted exposures, such as water and air pollutants (e.g., heavy metals), tobacco and alcohol use, infections (e.g., viruses and bacteria), ultraviolet or artificial light exposure, and dietary patterns, all of which are influenced by individual behaviors and local environmental conditions. The internal domain reflects the body’s biological responses to external exposures, including chronic inflammation, oxidative stress, metabolic dysregulation, immune activation, and aging processes. Together, these domains capture the dynamic and cumulative nature of exposures that shape disease susceptibility, including cancer risk, over time.

**Figure 2 cancers-17-02537-f002:**
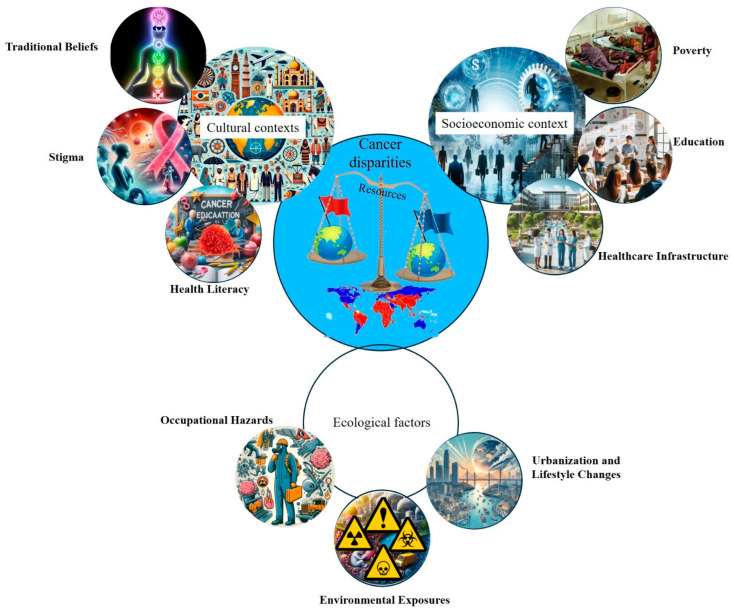
Cancer disparities in LMICs. This conceptual diagram highlights the diverse and interconnected cultural, socioeconomic, and environmental determinants of cancer disparities in LMICs. Using the exposome framework, it illustrates how cumulative exposure to environmental pollutants, infectious agents, lifestyle behaviors, and healthcare access barriers interact with systemic inequities to influence cancer incidence, progression, and outcomes. This figure underscores the need for integrated, context-specific strategies to address the multifactorial nature of cancer risk in resource-limited settings.

**Figure 3 cancers-17-02537-f003:**
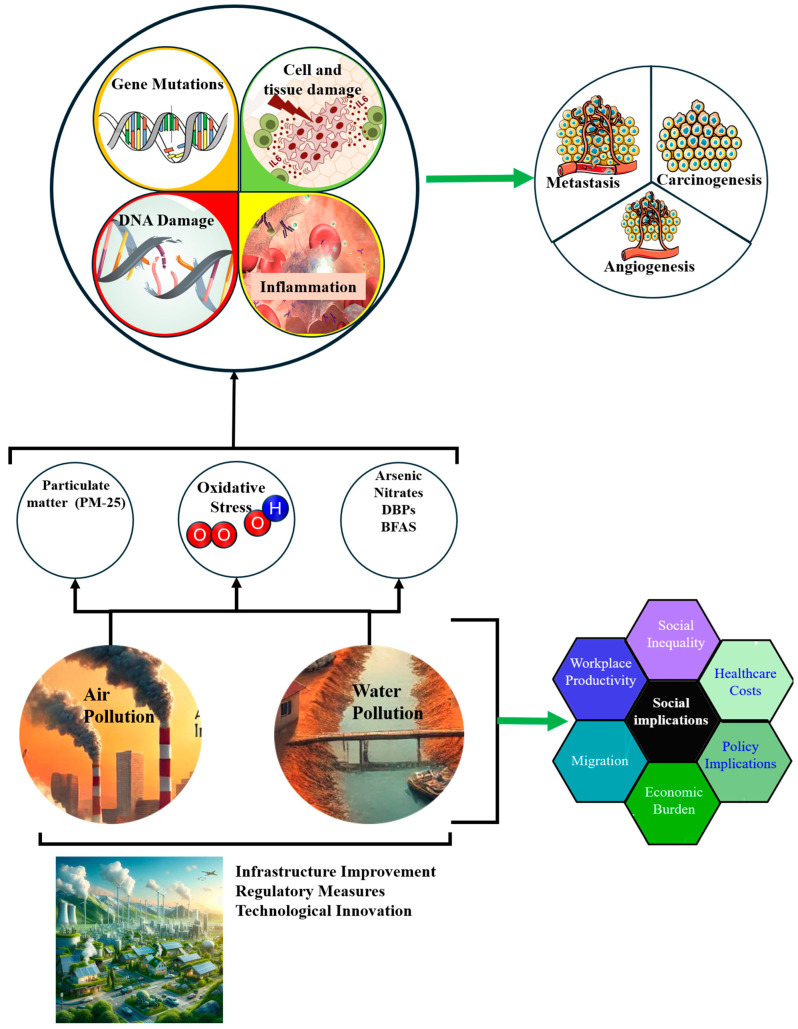
Environmental Health Issues in African LMICs. The diagram demonstrates the complex pathways through which air and water pollution (PM-25, arsenic, nitrates, DBPs, and BFAS) leads to oxidative stress, resulting in gene mutations, DNA damage, inflammation, and subsequent health issues such as metastasis, carcinogenesis, and angiogenesis. Social implications include increased healthcare costs, social inequality, and decreased workplace productivity. Efforts to mitigate these impacts include infrastructure improvements, regulatory measures, and technological innovations.

**Figure 4 cancers-17-02537-f004:**
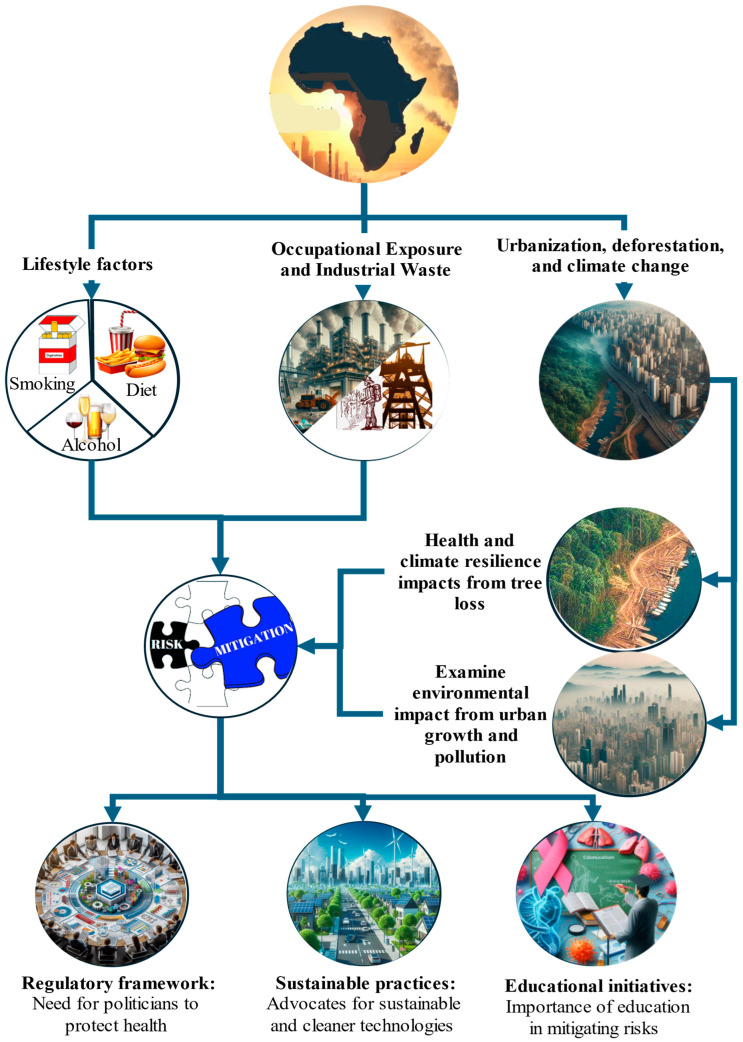
Occupational exposure and industrial waste in Africa. This figure highlights occupational risks, industrial waste management, regulatory frameworks, and educational initiatives, emphasizing their roles in mitigating health impacts and addressing cancer. This figure further highlights sustainable practices, the impacts of urbanization, and the effects of deforestation, emphasizing their roles in air pollution, climate resilience, and public health in African LMICs.

**Figure 5 cancers-17-02537-f005:**
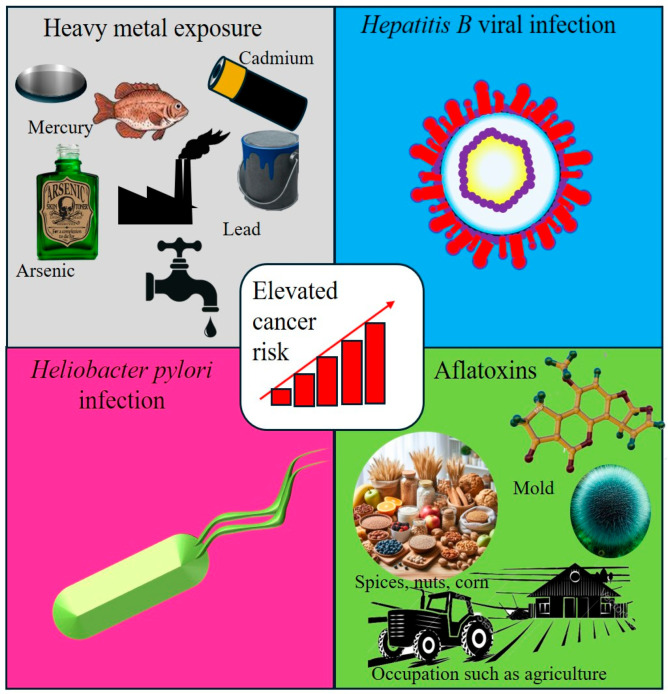
Interactions between infectious agents and environmental exposure. This figure illustrates the interplay between infectious agents and environmental exposure, both of which contribute to the risk of cancer. It highlights key carcinogenic factors, including heavy metals, Hepatitis B Virus (HBV), *Helicobacter pylori* (*H. pylori*), and aflatoxins, emphasizing their synergistic effects in promoting carcinogenesis through chronic inflammation, DNA damage, and immune modulation.

**Figure 6 cancers-17-02537-f006:**
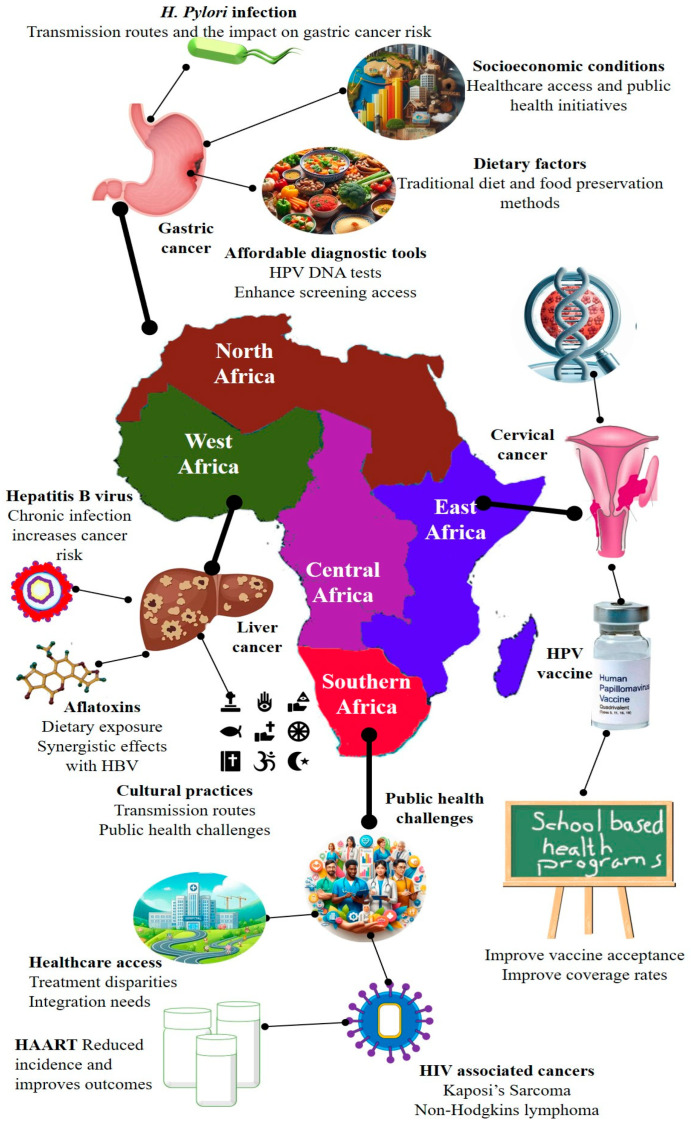
Regional cancer burden and associated risk factors in Africa. This map highlights the geographic distribution of predominant cancer types and their associated risk factors across African regions. In North Africa, the high prevalence of gastric cancer is primarily driven by *Helicobacter pylori* infection, dietary habits, and socioeconomic disparities. West and Central Africa bear a significant burden of liver cancer, largely attributed to chronic hepatitis B virus (HBV) infection and dietary aflatoxin exposure. In East Africa, cervical cancer is the most common malignancy among women and is closely linked to persistent human papillomavirus (HPV) infection and limited access to screening services. Southern Africa exhibits a high incidence of HIV-associated cancers, particularly Kaposi’s sarcoma and non-Hodgkin lymphoma, exacerbated by limited healthcare access and uneven antiretroviral therapy (ART) coverage. This figure underscores the urgent need for affordable diagnostics (e.g., HPV DNA testing), expanded vaccine coverage, and culturally sensitive, school-based health education to reduce cancer disparities across the African continent.

**Figure 7 cancers-17-02537-f007:**
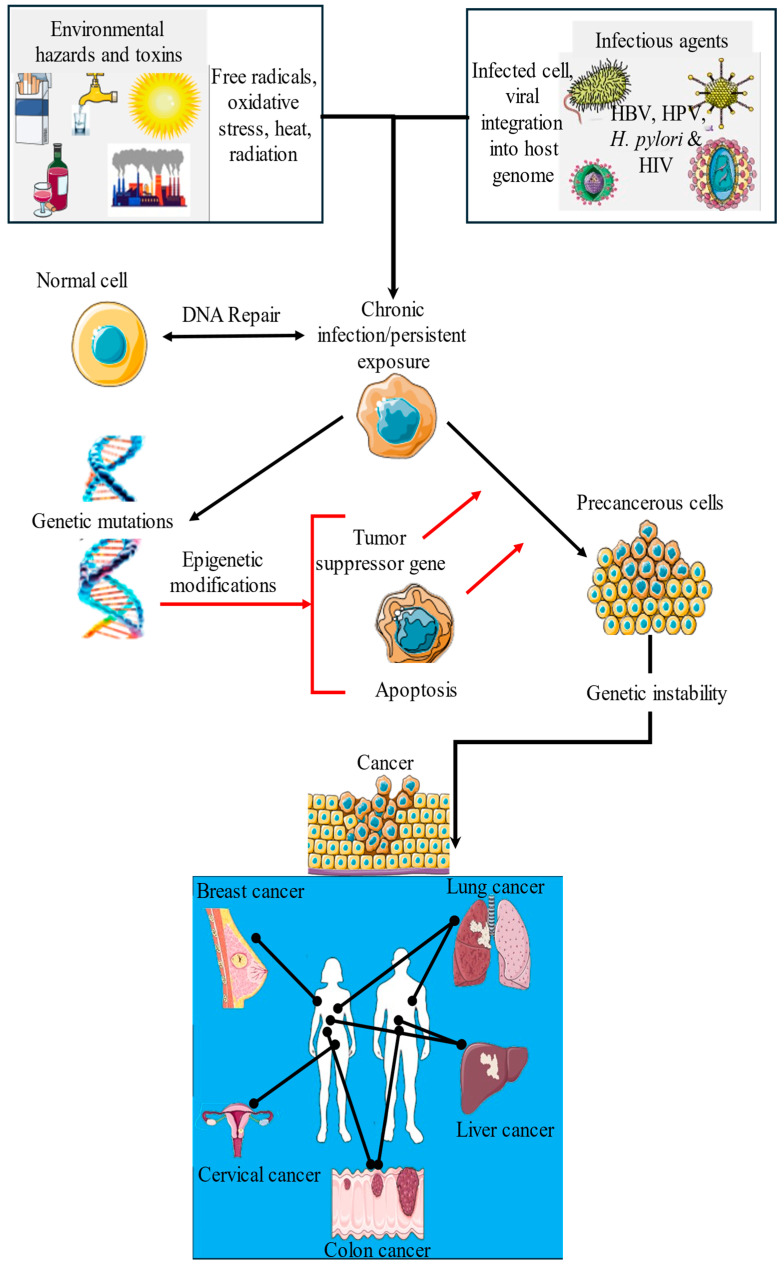
Synergistic interactions between environmental toxins and infectious agents in cancer development. This schematic illustrates how environmental exposures such as air pollutants, tobacco smoke, unhealthy diet, and alcohol consumption interact with infectious agents, such as HBV, HPV, *H. pylori*, and HIV, to drive the onset and progression of various cancers (e.g., lung, breast, colon, liver, and cervical). These combined exposures lead to genetic and epigenetic alterations, such as DNA damage, gene mutations, and disruption of gene expression. These molecular changes impair key cellular processes, including apoptosis, cell cycle control, and immune surveillance, thereby promoting cancer development. This figure highlights the complex and multifactorial etiology of cancer, particularly in high-exposure settings that are common in LMICs.

**Figure 8 cancers-17-02537-f008:**
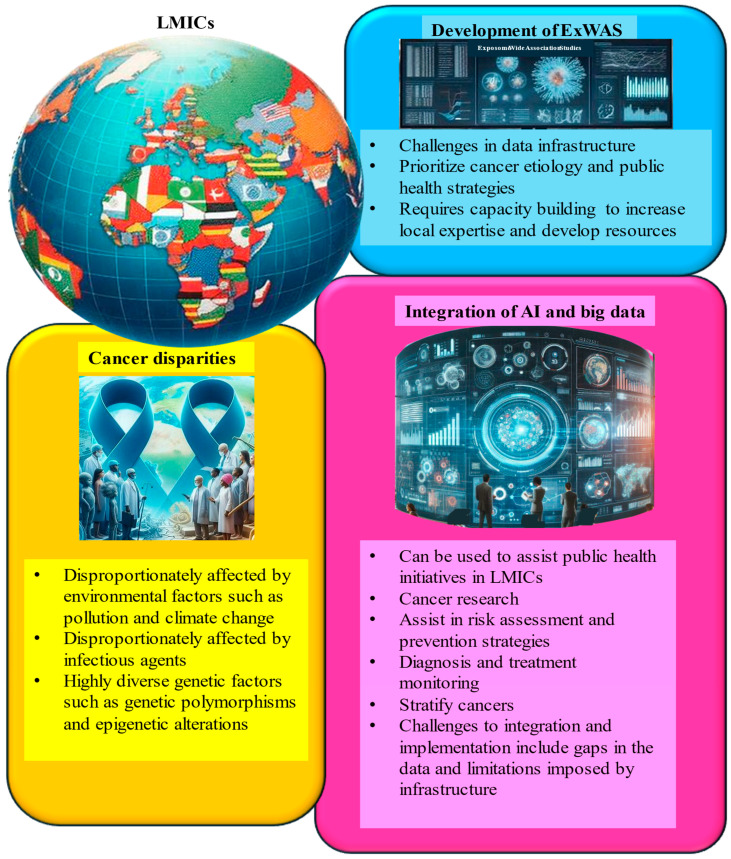
Integrated framework for reducing cancer disparities in low- and middle-income countries (LMICs) through exposome research and digital innovation. This schematic presents a multidisciplinary strategy to address cancer disparities in LMICs by combining exposome-wide association studies (ExWASs), digital health tools, and precision public health interventions. Core components include the identification of environmental and infectious risk factors through ExWASs, the use of artificial intelligence (AI) and big data analytics for population-level exposure mapping and risk prediction, and the development of localized capacity for implementation. The model emphasizes equitable access, community-driven data collection, and the integration of digital surveillance with targeted cancer prevention and control programs.

## Data Availability

Data sharing is not applicable to this review, as no new datasets were generated or analyzed.

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
