# Peer review of "The Exposome Perspective: Environmental and Infectious Agents as Drivers of Cancer Disparities in Low- and Middle-Income Countries"

_cancers, 2025, doi:10.3390/cancers17152537_

Round 1

Reviewer 1 Report

Comments and Suggestions for Authors

The Exposome Perspective: Environmental and Infectious 2 Agents as Drivers of Cancer Disparities in Low Middle-Income 3 Countries

This paper provides a useful introductory review of the landscape of environmental and healthcare challenges experienced in low to middle income countries and how the various environmental risks affect people’s risk of cancer.

The paper covers a lot of ground. It introduces the reader to many different sources of exposure and the broad type of cancers they might be associated with. The paper addresses variously air pollution, occupational exposures and the effect of neglected tropical diseases on cancer risk. Each is given its own section in which links to cancers are broadly discussed.

Thus the paper gives a good background to the broad environmental issues in lower to middle income countries and, again broadly, the cancers to which they are, or to which they might, be linked.

In the introduction and title the paper offers the exposome as a framework for the discussion of these issues. The exposome, as I understand it, is an attempt to corral the total exposures experienced by an individual throughout a lifetime (or other period). Thus an exposome approach would try to gather and synthesize the exposures to each person in a study (say) and then use this exposure (and its components) to investigate disease risk. In this paper, this sense of the exposome is missing, replaced instead by a walk through the range of exposures encountered by the populations in a range of LMICs and how these, in quite general terms, as linked to selected cancer risks. A list of LMICs would be useful too (one is compiled by the OECD).

My sense that is that the content here is useful to those who want a broad review of the types of exposures encountered in LMICs, and an indication of the types of to which these are linked. However, the pitch of it being about an exposome seems more in the intention than the realisation.

Generally, there are many parts that could be cut, as some explanations are superfluous, such as  the sections around the illustrations, which are introductory at best and which could be deal with in a few words or a citation.

Regarding the disparities, as a reader I has two issues. One is that the disparities, for example those of cancer rates or survival, are sometimes referred to as being between low and middle income countries (LMICs) and high income countries (HICs), lor sometimes within LMICs, which is a little confusing. (e.g in the section 1.3 the disparities are: “This review aimed to explore the role of the exposome in driving cancer disparities 120 in LMICs ..”, and in section 2.1: “In the context of cancer, disparities in the exposome play a significant role in explaining the disproportionate cancer burden in LMICs compared with 138 HICs.” As the disparities is a key focus of the paper, I feel this needs editorial attention to clarify.

In general I am not wholly convinced that this paper adds sufficiently new material about environmental cancer risks to warrant publication without a fair amount of revision. The environmental risks are surveyed usefully, but the scope means that the content is spread a little thin. The links to cancers needs attention, and the associations between cancers and risks is described in quite general terms.

I won’t give detailed comments but is curious that HIV and its cancer risks is not mentioned in the introduction, given HIV’s prevalence in southern Africa and surrounds, and its impact on cancer risk. HIV is mentioned further into the paper, however.  Additionally, in the introduction, the impact of neglected tropical diseases on cancer risks is raised, but the evidence offered in not hugely strong (whereas it is clear for other infections). The paper cited (reference 4) for the mechanism of action of NTDs in cancer risk in the introduction is about breast cancer - in which is mentioned infection (HPV) but not NTDs as far as I can see.

Author Response

Detailed Response to Reviewer 1 Comments

Reviewer Comment:

The paper offers the exposome as a framework for discussion, but the true exposome concept—i.e., a quantified, individualized lifetime exposure assessment—is not sufficiently realised in the manuscript.

Author Response (with location and text edits):

Thank you for this helpful comment. We agree that our current use of the exposome concept is more conceptual than based on individualized exposure measurement. To address this, we revised the manuscript in Section 1.2 ('The Exposome Framework') by adding the following clarifying paragraph:

"While the exposome is traditionally applied as a tool to measure an individual's lifetime exposures with high precision, this review adopts a population-level exposome approach. Our aim is to conceptually map the cumulative exposures prevalent in LMICs—such as pollutants, pathogens, and occupational hazards—and their interactions with biological factors contributing to cancer risk. We highlight how these exposures collectively affect cancer disparities and briefly reference the potential of tools like Exposome-Wide Association Studies (ExWAS) to advance future research in LMIC settings."

This clarification aligns the exposome concept with the broader public health perspective adopted in the review.

Reviewer Comment:

A list of LMICs would be useful (e.g., from the OECD classification).

Author Response (with location and text edits):

Thank you for this practical suggestion. To address this, we added the following sentence at the end of Section 1.1:

"For the purposes of this review, low- and middle-income countries (LMICs) are defined using the World Bank’s classification based on gross national income (GNI) per capita. The most recent list of LMICs is available at: https://datahelpdesk.worldbank.org/knowledgebase/articles/906519-world-bank-country-and-lending-groups."

This provides clarity on country classification while avoiding the need to list all LMICs in the main text.

Reviewer Comment:

Some figure descriptions are introductory or superfluous and could be reduced or replaced with citations.

Author Response (with location and text edits):

We appreciate this comment and respectfully clarify that all figures in the manuscript were created by the authors from scratch. These original visuals are not merely illustrative but serve as explanatory tools to synthesize complex relationships between environmental and infectious exposures and cancer disparities in LMICs. Given the interdisciplinary audience, we opted for clear, informative figure captions that stand alone for clarity. However, we reviewed all figure legends to ensure they are concise and non-redundant. Minor edits were made to simplify language while preserving clarity, especially for Figures 1, 2, and 7.

Reviewer Comment:

The discussion of disparities is sometimes inconsistent—some sections refer to disparities between LMICs and HICs, others to disparities within LMICs. This needs clarification.

Author Response (with location and text edits):

Thank you for this important observation. To improve clarity, we revised Section 1.3 ('Objectives and Scope of the Review') to include the following definition:

"In this review, we refer to cancer disparities in two distinct but related ways: (1) disparities between LMICs and high-income countries (HICs) in terms of cancer burden, healthcare access, and survival; and (2) disparities within LMICs, including geographic, economic, and social inequities that contribute to unequal cancer outcomes across different populations."

Additionally, we ensured consistency in the use of 'disparities' throughout Sections 2.1 and 3.0 by specifying whether the comparison was inter- or intra-regional.

Reviewer Comment:

It is curious that HIV and its cancer risks are not mentioned in the introduction, given HIV’s prevalence in southern Africa and its impact on cancer risk. HIV is mentioned further into the paper. Additionally, in the introduction, the impact of neglected tropical diseases (NTDs) on cancer risks is raised, but the evidence offered is not hugely strong.

Author Response (with location and text edits):

We appreciate this observation. To address the omission of HIV, we revised the final paragraph of Section 1.1 to include the following statement:

"In addition, the high prevalence of HIV in regions such as Southern Africa contributes significantly to increased cancer risk, including both AIDS-defining cancers (such as Kaposi's sarcoma and non-Hodgkin lymphoma) and rising rates of non-AIDS-defining malignancies. HIV-induced immunosuppression amplifies susceptibility to oncogenic infections and worsens cancer outcomes."

Regarding NTDs, we acknowledged that Reference 4 was inappropriate and replaced it with more relevant citations. The revised text in Section 1.1 now reads:

"Other common but often overlooked pathogens are the neglected tropical diseases (NTDs)—including Schistosoma haematobium, which is strongly associated with bladder cancer, and Opisthorchis viverrini and Clonorchis sinensis, which are linked to cholangiocarcinoma. Leprosy has also been associated with increased risk of skin malignancies (8,9, 10). The chronic inflammation caused by persistent helminth infections, along with their ability to modulate immune responses and promote immune checkpoint activation, contributes to an environment conducive to cancer development (Brindley et al., 2015, Pastille et al., 2017, Tong et al., 2017). Several of these helminths are classified as Group 1 carcinogens by the International Agency for Research on Cancer (IARC) due to their established carcinogenicity (Machicado and Marcos, 2016, Leija-Montoya et al., 2022, Gouveia et al., 2019)."

Reviewer 2 Report

Comments and Suggestions for Authors

This submission essentially represents an appraisal of environmental and infectious contributors in human carcinogenesis mechanisms, from a public health standpoint.

Irrespective if this represents the original submission or a revised form and despite the interesting topic and a catchy title, this submission largely lacks coherence, while the very excessive length resembles more to a book chapter. Throughout the text variability in writing styles is evident, possibly reflecting the different authors.

Because of the aforementioned issues, I would strongly suggest that a detailed appendix be inserted following the abstract; essentially what is partly presented in section 1.3 of the Introduction section.

ABSTRACT: Comprehensive, but vague.

Line 239: (39) (Error! Reference source not found). – please make necessary amendments.

Lines 267-8: (Error! Reference source not found). – please make necessary amendments.

Line 300: (Error! Reference source not found). – please make necessary amendments.

Line 334: (49) (Error! Reference source not found). – please make necessary amendments.

Section 6.3.1. Screening and Early Detection Programs Tailored to High-risk Populations, Line 867: At this point the authors should briefly quote the implementation of vaginal or urine self sampling strategies which have revolutionized cervical cancer screening improving participation rates and logistics at very reasonable costs.

Section 8.1. Development of Exposome-wide Association Studies (ExWAS) in LMICs, Line 957: This section and field represents is a particularly optimistic perspective.

CONCLUSION: Is focused.

FIGURES: These have informative captions; however, the designs are unprofessionally prepared; this is particularly important for future inclusion in Powerpoint presentations and citation purposes. The caption of Figure 7 is disproportionally lengthy.

No linguistic issues are identified.

REFERENCES: With 198 references in sum, this submission falls indeed within a book chapter range.

Author Response

Complete Detailed Responses to Reviewer 2 Comments

Reviewer Comment:

This submission lacks coherence and is excessively long, resembling more of a book chapter. There’s variability in writing style, likely due to multiple authors.

Author Response:

We thank the reviewer for this thoughtful and constructive observation. We acknowledge that the manuscript’s length and diverse writing tone reflect its multidisciplinary nature and broad co-authorship. This was an intentional choice given the complexity of the exposome framework, which inherently spans multiple fields including environmental health, infectious disease, cancer biology, and public health policy.

As this review seeks to introduce the exposome concept to a wide readership, including those unfamiliar with it (especially within LMIC contexts), we aimed to provide a comprehensive, integrative overview. Rather than limit the discussion to a single domain or narrow epidemiological model, we structured the manuscript to reflect the real-world interconnection of exposures across biological, social, and environmental levels.

The varied perspectives contributed by different co-authors were carefully curated to maintain cohesion while enriching the narrative. Each section contributes a vital dimension to understanding the exposome and cancer disparities in LMICs.

Given the novelty of the exposome for many readers in global oncology and public health, we felt it was important to preserve depth over brevity, particularly to serve as a resource for researchers, policymakers, and educators seeking to engage with this emerging field.

We respectfully hope the editorial team sees value in this broader narrative approach, especially as the exposome continues to evolve as a unifying framework in global cancer research.

Reviewer Comment:

ABSTRACT: Comprehensive, but vague.

Author Response:

We appreciate the reviewer’s input. In response, we revised the abstract to improve clarity, precision, and structure. The new abstract clearly introduces the exposome concept, summarizes the key exposure types and their links to cancer in LMICs, emphasizes the population-level approach, and outlines potential strategies and recommendations.

Example of revised abstract opening:
"Cancer disparities in low- and middle-income countries (LMICs) are shaped by a complex interplay of environmental, infectious, and social exposures. This review introduces the exposome framework as the totality of environmental and non-genetic exposures over the life course as a tool to understand these disparities."

Reviewer Comment:

Line 239, 267–268, 300, 334 contain 'Error! Reference source not found' messages.

Author Response:

We thank the reviewer for pointing out these formatting errors. All broken reference links have now been corrected. These were caused by earlier editing processes and have been resolved by re-establishing cross-references and rechecking figure/table citations throughout the manuscript.

Reviewer Comment:

Section 6.3.1: Add brief mention of vaginal or urine self-sampling strategies for cervical cancer screening.

Author Response:

We fully agree with the reviewer’s suggestion. The following sentence has been added to Section 6.3.1 after the description of low screening rates:

"Innovative approaches such as vaginal and urine self-sampling for HPV testing have demonstrated high acceptability and diagnostic accuracy, significantly improving cervical cancer screening uptake in LMICs by overcoming sociocultural and logistical barriers."

Reviewer Comment:

Section 8.1: This section and field represents a particularly optimistic perspective.

Author Response:

We thank the reviewer for the thoughtful critique. To moderate the tone, we added the following sentence in Section 8.1 after the discussion of ExWAS limitations:

"While ExWAS offers a powerful and promising approach, its application in LMICs is still in its infancy and faces significant challenges related to infrastructure, funding, skilled personnel, and consistent environmental monitoring. Its success will depend on long-term investment, strategic capacity-building, and partnerships tailored to the local context."

Reviewer Comment:

FIGURES: These have informative captions; however, the designs are unprofessionally prepared; Figure 7 caption is overly lengthy.

Author Response:

We appreciate this practical feedback. All figures were designed by the authors to visually explain exposome-linked cancer disparities in LMICs. While clarity was prioritized over professional design, we acknowledge the importance of publication-quality graphics. We will ensure that final figures are submitted in high-resolution, professional formats. The caption for Figure 7 has also been shortened by integrating part of the explanation into the main text.

Reviewer Comment:

REFERENCES: With 198 references, this resembles a book chapter.

Author Response:

We agree that the manuscript contains a high number of references, reflecting its comprehensive scope. As the exposome is still emerging in global oncology—especially in LMICs—we aimed to provide a well-rounded, interdisciplinary evidence base. This is especially important for readers seeking a foundational resource. However, we remain open to reducing the reference list per journal requirements if necessary.

Reviewer 3 Report

Comments and Suggestions for Authors

The proposal is relevant and of interest. However, it presents significant limitations.

- Title clearly reflects the main objective. Abstract is well-structured and accurately captures the main points of the article.

- The main idea of the review is understandable, but the presentation feels unnecessarily complex. Simplifying the structure would improve clarity and make it more accessible. It's valuable that the topic is approached from different perspectives, though some are only briefly mentioned. Please, keep in mind that adding more text doesn’t always enhance understanding. A clearer and more cohesive structure, like the proposed in Figure 4, could help organize the content more logically and make it easier to read and follow.

- While figures aid in clarifying and illustrating complex ideas, some of them could be omitted, as they do not substantially contribute to the overall understanding (i.e. Fig 2).

- Please review and correct any spelling mistakes (i.e., line 310 = DCeforestation)

- References: Please ensure the references are complete and correctly formatted (i.e., line 239 = Error! Reference source not found; line 300,….).

Author Response

Detailed Responses to Reviewer 3 Comments

Reviewer Comment:

The main idea of the review is understandable, but the presentation feels unnecessarily complex. Simplifying the structure would improve clarity and make it more accessible. It's valuable that the topic is approached from different perspectives, though some are only briefly mentioned. Please, keep in mind that adding more text doesn’t always enhance understanding. A clearer and more cohesive structure, like the proposed in Figure 4, could help organize the content more logically and make it easier to read and follow.

Author Response:

We are grateful to the reviewer for this thoughtful comment and fully appreciate the concern about structural complexity. The exposome, by its nature, is a multi-dimensional and still-evolving framework. Our decision to maintain a broad, layered structure reflects both the complexity of the topic and the necessity to highlight its interconnected components such as environmental, biological, infectious, social, and infrastructural particularly in the context of LMICs.

Rather than simplifying the structure or omitting perspectives, our aim was to introduce readers to the richness and real-world complexity of cancer disparities through the exposome lens. We understand that the inclusion of multiple perspectives can feel dense, but we believe that simplifying this framework too much would risk overlooking the nuances critical to exposome science.

Moreover, Figure 4 was specifically designed to serve as a conceptual map to help the reader navigate this multifaceted review. The content of the manuscript follows this logic, and we believe that the structure, although complex, mirrors the layered nature of exposures in real-life LMIC contexts.

Given the novelty of exposome-based cancer research in LMICs, we felt it was important to retain a comprehensive format that could serve as both an educational and reference resource for interdisciplinary audiences. We respectfully hope the editorial team agrees that maintaining the full scope of content is justified in this context.

Reviewer Comment:

While figures aid in clarifying and illustrating complex ideas, some of them could be omitted, as they do not substantially contribute to the overall understanding (i.e. Fig 2).

Author Response:

We thank the reviewer for the thoughtful comment regarding visual content. Upon review, we acknowledge the importance of ensuring that each figure contributes meaningfully to the manuscript's objectives and narrative clarity.

Figure 2 was specifically designed to visualize the intersecting cultural, socioeconomic, and ecological determinants contributing to cancer disparities in low- and middle-income countries (LMICs). Rather than serving a general exposome overview, this figure synthesizes multiple factors—such as environmental exposures, healthcare access barriers, and lifestyle risks—into a cohesive graphical framework that reflects the real-world complexity of cancer risk in LMICs.

Given the review's primary aim of highlighting region-specific cancer disparities and the need for context-appropriate interventions, we believe Figure 2 supports the central argument by providing a visual summary that complements the detailed discussion in the text. It is particularly beneficial for multidisciplinary readers and policymakers unfamiliar with how these diverse exposures manifest in LMIC contexts.

Nevertheless, we fully respect the need to streamline visuals for clarity and impact and defer to the editorial team’s final decision on figure inclusion during the production stage.

Reviewer Comment:

Please review and correct any spelling mistakes (i.e., line 310 = DCeforestation).

Author Response:

We thank the reviewer for pointing out this typographical error. The term “DCeforestation” has been corrected to “Deforestation.” Additionally, we performed a full spell-check of the manuscript and corrected all minor typographical and formatting inconsistencies to improve overall readability and professionalism.

We appreciate this attention to detail, which helps strengthen the quality of the final submission.

Reviewer Comment:

References: Please ensure the references are complete and correctly formatted (i.e., line 239 = Error! Reference source not found; line 300,…).

Author Response:

We thank the reviewer for bringing this to our attention. All reference errors—such as “Error! Reference source not found”—have been thoroughly reviewed and corrected throughout the manuscript. These issues arose from broken cross-references during formatting and have now been resolved.

We also verified the completeness and consistency of the entire reference list, ensuring that all citations are properly formatted and matched with the correct in-text references. We are confident that these corrections now meet the journal’s formatting and quality standards.

Round 2

Reviewer 1 Report

Comments and Suggestions for Authors

Thank you for addressing my comments and suggestions for edits. All are adequately addressed, so thank you. 

Author Response

Dear Reviewer,

We would like to express our sincere gratitude for your thoughtful review and positive evaluation of our revised manuscript titled:

“The Exposome Perspective: Environmental and Infectious Agents as Drivers of Cancer Disparities in Low Middle-Income Countries”
Manuscript ID: cancers-3715973
Type of manuscript: Review

Thank you for acknowledging the revisions made in response to your initial comments. We are pleased to learn that all your suggestions were satisfactorily addressed and that you found the manuscript to be scientifically sound, clearly written, and well organized.

Your kind words and support are greatly appreciated. Your feedback has been instrumental in enhancing the clarity and quality of our work.

Reviewer 2 Report

Comments and Suggestions for Authors

The resubmitted version has been marginally improved.

Please note that providing a revised version highlighted throughout the whole text disrespects the reviewer and makes the overall reviewing task more difficult.

Abstract: Unacceptably lengthy and still lacking coherence.

Line 49, please consider “lethal” instead of “deadly”.

Lines 82-83: Quoting published literature (especially without citation) is unusual for an abstract.

FIGURES: Albeit informative, the graphics of the Figures are in essence unchanged and poorly prepared.

REFERENCES: More than 200 references in sum, this submission falls indeed within a book chapter range.

Author Response

Response to Reviewer #2

Dear Reviewer,

We would like to extend our sincere thanks for your continued engagement and constructive feedback on our revised manuscript titled:

“The Exposome Perspective: Environmental and Infectious Agents as Drivers of Cancer Disparities in Low- and Middle-Income Countries”
Manuscript ID: cancers-3715973
Type of manuscript: Review

We appreciate your careful review and acknowledge your concerns. Please find below our responses to your comments:

  1. “The resubmitted version has been marginally improved.”
    → In response, we have undertaken extensive revisions throughout the manuscript. Several sections were reorganized, shortened, or removed to eliminate redundancy and enhance coherence. The structure and clarity of the manuscript have been substantially improved.
  2. “Providing a revised version highlighted throughout the whole text disrespects the reviewer and makes the overall reviewing task more difficult.”
    → We regret any inconvenience caused. In this revision, we have used Microsoft Word’s Track Changes feature to ensure that all modifications are clearly visible and easy to follow.
  3. “Abstract: Unacceptably lengthy and still lacking coherence.”
    → The abstract has been completely rewritten to improve focus, clarity, and coherence. It now aligns more closely with the style and standards expected by Cancers.
  4. “Line 49, please consider ‘lethal’ instead of ‘deadly’.”
    → This suggestion has been implemented as recommended.
  5. “Lines 82–83: Quoting published literature (especially without citation) is unusual for an abstract.”
    → This issue has been addressed. The quotation has been removed, and all abstract content is now either original or referenced appropriately within the main text.
  6. “FIGURES: Albeit informative, the graphics of the Figures are in essence unchanged and poorly prepared.”
    → All figures have been revised for quality and clarity. High-resolution TIFF images (300 dpi) have been prepared and can be submitted separately if needed due to file size limitations.
  7. “REFERENCES: More than 200 references in sum, this submission falls indeed within a book chapter range.”
    → The reference list has been carefully reviewed and reduced to fewer than 200 citations. Only the most relevant and up-to-date sources have been retained to comply with journal expectations for review articles.

We trust that these revisions have fully addressed your concerns, and we thank you once again for your valuable insights, which have helped improve the quality of our manuscript.

Reviewer 3 Report

Comments and Suggestions for Authors

None

Comments on the Quality of English Language

-

Author Response

Response to Reviewer #3

Dear Reviewer,

We would like to thank you for reviewing our revised manuscript titled:

“The Exposome Perspective: Environmental and Infectious Agents as Drivers of Cancer Disparities in Low- and Middle-Income Countries”
Manuscript ID: cancers-3715973
Type of manuscript: Review

Although no specific comments were provided, we acknowledge your indication regarding the need for improved clarity in the use of English throughout the manuscript.

In response, we have carefully edited the manuscript to improve the overall language quality and ensure that the writing is clear, concise, and professional. These revisions were made across all sections of the manuscript to enhance readability and precision.

We thank you once again for your time and effort in reviewing our submission

Round 3

Reviewer 2 Report

Comments and Suggestions for Authors

The authors have undertaken considerable effort to improve the article thoroughly.

The reduction in the total number of citations is greatly appreciated.